# Genome-Wide Associations with Resistance to Bipolaris Leaf Spot (*Bipolaris oryzae* (Breda de Haan) Shoemaker) in a Northern Switchgrass Population (*Panicum virgatum* L.)

**DOI:** 10.3390/plants11101362

**Published:** 2022-05-20

**Authors:** Kittikun Songsomboon, Ryan Crawford, Jamie Crawford, Julie Hansen, Jaime Cummings, Neil Mattson, Gary C. Bergstrom, Donald R. Viands

**Affiliations:** 1Section of Plant Breeding and Genetics, School of Integrative Plant Science, Cornell University, Ithaca, NY 14853, USA; rvc3@cornell.edu (R.C.); jln15@cornell.edu (J.C.); jlh17@cornell.edu (J.H.); drv3@cornell.edu (D.R.V.); 2Syngenta, Trumansburg, NY 14886, USA; jaime.cummings@syngenta.com; 3Section of Horticulture, School of Integrative Plant Science, Cornell University, Ithaca, NY 14853, USA; nsm47@cornell.edu; 4Section of Plant Pathology and Plant-Microbe Biology, School of Integrative Plant Science, Cornell University, Ithaca, NY 14853, USA; gcb3@cornell.edu

**Keywords:** disease resistance, Bipolaris leaf spot, switchgrass, genome-wide association

## Abstract

Switchgrass (*Panicum virgatum* L.), a northern native perennial grass, suffers from yield reduction from Bipolaris leaf spot caused by *Bipolaris oryzae* (Breda de Haan) Shoemaker. This study aimed to determine the resistant populations via multiple phenotyping approaches and identify potential resistance genes from genome-wide association studies (GWAS) in the switchgrass northern association panel. The disease resistance was evaluated from both natural (field evaluations in Ithaca, New York and Phillipsburg, Philadelphia) and artificial inoculations (detached leaf and leaf disk assays). The most resistant populations based on a combination of three phenotyping approaches—detached leaf, leaf disk, and mean from two locations—were ‘SW788’, ‘SW806’, ‘SW802’, ‘SW793’, ‘SW781’, ‘SW797’, ‘SW798’, ‘SW803’, ‘SW795’, ‘SW805’. The GWAS from the association panel showed 27 significant SNPs on 12 chromosomes: 1K, 2K, 2N, 3K, 3N, 4N, 5K, 5N, 6N, 7K, 7N, and 9N. These markers accumulatively explained the phenotypic variance of the resistance ranging from 3.28 to 26.52%. Within linkage disequilibrium of 20 kb, these SNP markers linked with the potential resistance genes included the genes encoding for NBS-LRR, PPR, cell-wall related proteins, homeostatic proteins, anti-apoptotic proteins, and ABC transporter.

## 1. Introduction

Switchgrass (*Panicum virgatum* L.) is a perennial biomass crop native to North America. Biomass and other agronomic traits have been the focus of breeding programs [1,2,3,4]. Although many diseases have been reported to cause deleterious effects on yield, research on disease resistance, especially breeding for the resistance, is scarce. Only 47 of the 2328 research articles directly related to diseases in switchgrass [5]. *Bipolaris oryzae* (Breda de Haan) Shoemaker (teleomorph: *Cochliobolus miyabeanus*) is one of the major fungi causing Bipolaris leaf spot (BLS) in switchgrass that can reduce biomass by 70% [6]. To manage the disease, breeding is one of the most economical approaches [7].

The natural distribution of switchgrass stretches latitudinally across North America east of the Rocky Mountains. By simple morphological differentiation, switchgrass can be separated into two groups of upland and lowland ecotypes. Upland ecotypes are well adapted to higher latitudes and have higher drought tolerance, whereas lowland ecotypes provide higher yield and require more water [8]. The more in-depth genetic diversity was revealed by the Network-Based single nucleotide polymorphism (SNP) discovery protocol [9]. It suggested the differentiation of switchgrass groups by isolation-by-ploidy, the migration from south to north, and the incidence of tetraploid upland from octaploid upland. Such a large diversity of switchgrass populations can provide a source of disease resistance [10].

Plant-pathogen interactions are complicated. The basal defense is initiated when conserved molecular signatures of pathogens called pathogen-associated molecular patterns (PAMPs) are recognized by plant pattern recognition receptors (PRRs). Such a recognition activates PAMP-triggered immunity (PTI). Successful pathogens can suppress PTI by secreting virulent effector proteins. These effectors then trigger the second line of defense called effector-triggered immunity (ETI), mediated by resistance R genes [11]. In a biotrophic pathogen, a gene-for-gene interaction between resistance R genes of plant and avirulent (Avr) genes of the pathogen results in a hypersensitive response leading to local programmed cell death and stopping colonization conferring resistance. Since *B. oryzae* is a necrotrophic pathogen [12], the interaction between switchgrass and the fungus was modeled potentially as an inverse gene-for-gene model [13]. In general, the pathogen secretes necrotrophic effectors as host-selective toxins (HST), such as SnTox1 from *Stagonospora nodorum*, T-toxin from *Cochliobolus heterostrophus*, and HC-toxin from *C. carbonum* [14,15,16]. The toxins interact with host sensitivity genes, resulting in a compatible susceptible interaction to trigger host cell death [17]. Such an interaction is known as effector-triggered susceptibility (ETS). Although crude extract from *B. oryzae* was proposed to contain HST [18], the toxin has never been characterized. Instead, according to a comparative genome study [19] and screening on a wide range of hosts [20,21], *B. oryzae* does not produce any HST but produces ophiobolin A and B as a non-host-selective toxin [22], triggering many pathways such as reactive oxygen species (ROS) detoxification, protein phosphorylation, and ethylene production [23,24].

Such an ETS interaction suggested that susceptible cultivars can rapidly screen for pathogens carrying the necrotrophic effectors [25]. Breeding for the improvement of resistance to BLS can be done by eliminating susceptible alleles from the population. However, the recurrent phenotypic selection for the resistance in upland switchgrass cultivars ‘Shelter’ and ‘Cave-in-Rock’ for two cycles of selection did not improve resistance to BLS [26]. The screening was done in a seedling stage; therefore, future breeding for the resistance with different screening methods, in different growth stages, and in bigger populations was suggested. Identification of resistant genotypes or populations is still in need.

To accelerate breeding for disease resistance, genomics-assisted breeding is an important approach [27]. Basically, it begins with gene identification, isolation, cloning, functional characterization, validation, and utilization. There are two main approaches to identify resistance genes in a diverse population: linkage mapping and genome-wide association studies (GWAS). Although GWAS cannot confirm the causal polymorphism, it depends on linkage disequilibrium (LD) that potentially links between the markers and the causal polymorphisms. The diversity panel contains high allelic diversity and ancestral recombination events resulting in a finer resolution than linkage mapping [28]. The technique has been used to dissect flowering time in switchgrass [29]. Although GWAS has never been used to dissect disease resistance in switchgrass, it has successfully dissected resistance to leaf rust caused by *Puccinia triticina* Eriks., tan spot caused by *Pyrenophora tritici-repentis* (Die.) Shoemaker and stripe rust caused by *Puccinia striiformis* in wheat (*Triticum aestivum* L.) [25]. Despite no resistance genes to BLS identified in switchgrass, there were 13 SNP markers from quantitative trait loci (QTLs) linked with the resistance in rice [30,31]. This study will provide the first dissection of the resistance to BLS in switchgrass.

The objectives of this study were (1) to determine genotypes from the northern switchgrass association panel that can be candidates for resistance to BLS, (2) to dissect the resistance to BLS in the association panel via GWAS, and (3) to explore the genes linked to the significant markers from the GWAS to determine potential candidate genes for resistance to BLS.

## 2. Results

### 2.1. Phenotyping and Correlation between Traits

The association panel of 478 genotypes from 66 populations was evaluated for the severity of BLS from the mean of Bipolaris lesion percentage from detached leaf assay by vision (DTVI), by image analysis (DTIA), the severity from the leaf disk assay by vision (DSVI), by image analysis (DSIA), mean of two locations (Ithaca, New York and Phillipsburg, Philadelphia), the highest scores between two locations (MTL), mean in New York (NY), the highest score in NY (MNY), mean in Philadelphia (PA), and highest score in PA (MPA) based on population (rank). ‘Population’ in this study meant a seed source with a specific origin and, in some cases, breeding history. ‘Genotype’ meant the individual plants propagated from populations.

The results from the detached leaf showed that ‘ECS.6’ had the lowest severity and ‘SW115’ had the highest severity among populations (Appendix A). When comparing among genotypes, ‘SW788.05’ showed the lowest severity, whereas ‘SW63.05’ showed the highest severity (Appendix A). The result from the leaf disk assay showed that ‘SW803’ showed the lowest severity, and ‘SW38’ showed the highest severity. In the genotype-based comparison, ‘High Tide.02’ showed the lowest severity. Eighty-one genotypes showed the highest severity. In artificial inoculations, both detached leaf and leaf disk assays suggested that the lowland ecotypes (DTVI 43%, DTIA 21%, DSVI 43%, and DSIA 70%) were significantly more resistant than upland ecotypes (DTVI 61%, DTIA 39%, DSVI 67%, and DSIA 89%) (*p*-value < 0.05) (Appendix A).

In the field evaluation based on the mean of the severity score (0 to 5) from two locations, ‘Shelter’ showed the lowest severity (0.44) and ‘SW787’ showed the highest severity. However, when considering each location, different populations performed differently. In comparison between NY and PA, Wilcoxson’s rank test of the severity showed significant difference between the two locations (*p*-value = 0.042). The re-ranking incidence across locations indicates the significant GxE effect [32]. In NY, SW803 showed the lowest severity (0.22), and ‘SW787’ showed the highest severity (3.0). Whereas in PA, ‘Shelter’ showed the lowest severity (0.33), and ‘Pathfinder’ showed the highest severity (2.9). Between the two locations, ‘SW123’, ‘SW33’, ‘SW793’, ‘SW781’, ‘High Tide’, and ‘Timber’ had the lowest MTL at 3 (resistant), and 41 populations had the highest MTL at 5 (susceptible). In NY, ‘SW123’, ‘SW128’ and ‘ECS.6’ had the lowest MNY at 2, and 32 populations had the highest MNY at 5. In PA, ‘SW115’, ‘SW802’, ‘SW31’, and ‘SW43’ had the lowest MPA at 2, and 19 populations had the highest MPA at 5. Although each phenotyping approach cannot determine a single resistant population, a ranking of detached leaf, leaf disk, and mean from two locations showed that ‘SW788’, ‘SW806’, ‘SW802’, ‘SW793’, ‘SW781’, ‘SW797’, ‘SW798’, ‘SW803’, ‘SW795’, ‘SW805’ were always in the lowest severity group and can be the source of resistance (Appendix A). When comparing ecotypes, the field evaluation of means between two locations showed a trend of more resistance in lowland than upland ecotypes significantly (Appendix A). The difference between lowland and upland ecotypes also occurred in NY and PA. The severities based on genotypes in each phenotyping approach were listed (Appendix A).

Broad-sense heritabilities (H^2^) in different groups of genotypes were similar across groups with some variations (Table 1). For example, H^2^ from DSIA in upland was only 0.14 while other groups had around 0.40. Although the severity from different phenotyping approaches in different groups had significantly moderate to high H^2^, H^2^ of severity from two locations was zero. This was supported by the significantly different mean of severity from two locations, NY and PA (Appendix A and Wilcoxson *p*-value = 0.042). The different rankings did not only show in the field evaluation, but the other phenotyping approaches also gave different rankings of resistance from DTIA, DSIA, and mean from two locations and correlation among the traits (Appendix A and Wilcoxson’s *p*-value < 2.2 × 10^−16^). For example, from 478 genotypes, ‘KY1625_07’ ranked 27th from DTIA but ranked 329th in DSIA and 100th in mean from two locations. Such differences led to a low correlation among approaches. The r^2^ between DTIA and DSIA was only 0.1 and between DTIA and the mean of two locations (TL) was 0.03 (Figure 1). In contrast to these low correlations, there were high correlations in DTVI-DTIA (r^2^ = 0.84) and DSVI-DSIA (r^2^ = 0.85). There was no correlation between BLS resistance and other agronomic or biomass quality traits (Appendix A).

Additionally, Qst-Fst comparisons were used to investigate whether the mean difference of each phenotype among populations is greater than the expected mean difference under genetic drift. None of the traits (DTVI, DTIA, DSVI, and DSIA) has Qst-Fst significantly more than zero (Appendix A), showing no positive selection for resistance to BLS in populations.

### 2.2. Linkage Disequilibrium, Principal Component Analysis and Admixture

To determine the intervals that were potentially linked to the significant markers from GWAS, linkage disequilibrium (LD) decay was estimated by plotting the allele frequency correlations (R2) against the physical distance in base pairs (Figure 2). In the full set of 478 genotypes, the LD decayed sharply within 20 kb and reached a plateau background within 50 kb. Therefore, in this study, we focused on genes within the 20 kb from the significant SNPs.

Based on the principal component (PC) analysis and admixture, there was a population structure in this association panel (Figure 3a). From PC1 and PC2, upland and lowland ecotypes separated from each group. Moreover, within the lowland ecotype, the latitude of the lowland can be distinguished to lowland north and south groups. The separation of the upland ecotype can also be noticed in PC1 and PC3 that the upland north was grouped apart from the other upland. Besides ecotypes, ploidy levels can be differentiated into groups. Both lowland north and south, and upland north genotypes were tetraploid (4X), while the upland east and upland west genotypes were octaploid (8X). In total, PC1, PC2, and PC3 explained the variance due to the population of 49.35%. The admixture (Figure 3b) also showed the distinction among five gene pools, including lowland North, lowland South, upland East, upland North, and upland West.

### 2.3. Genome-Wide Association and Candidate BLS Resistance Genes

We conducted GWAS with five subsets of genotypes, including 478, 4X, 8X, lowland, and upland. Most of the single phenotypes did not show significant Manhattan peaks or theoretically expected QQ-plots. To improve the association analysis, multiple combinations of BLS severity approaches were implemented. The full set of 478 genotypes conducted the GWAS with the combination of three traits of DTVI, DTIA, and DSIA, which showed 14 significant peaks over the FDR threshold (Figure 4a and Table 2).

To compare with reported BLS resistance genes from QTLs in rice [30], 13 SNPs on three rice chromosomes within the LD of 200 kb were examined (Appendix A). There were 278 genes linked to these markers. Among them, 172 genes were annotated with 134 biological functions. The most frequent biological functions were Leucine-rich repeat family protein (6), peroxidase precursor (6), transposon protein (6), nucleotide-binding sites–leucine rice repeats (NBS-LRRs) type disease resistance protein (5), MYB family transcription factor (3), dirigent (2), pentatricopeptide repeat (PPR) domain (2), fasciclin domain-containing protein (2), flavin monooxygenase (2), and heavy metal associated domain (2). There were 151 genes from 278 genes that showed 16.7 hits on average from BLAST against the *P. virgatum* genome v.4.1 (Appendix A). The most hits of 207 were on Chr07N. The fewest hits of 58 were on Chr04N.

In a tetraploid group, the two-approached combination of DSVI and MNY provided three significant peaks with the expected QQ-plot (Figure 4b). Interestingly, one peak of SNP Chr07N_61547213 on chromosome 7N from tetraploid group overlapped with the same peak in GWAS from the 478 group. In the octaploid group, the two-approached combination of DTVI-DSIA showed one significant peak on chromosome 9N (Figure 4c). In the lowland group, the best GWAS was from the single trait mean of two locations yielding eight peaks (Figure 4d). In the upland group, the combination of DTIA-MTL gave the two most significant peaks on chromosome 1k and 7K (Figure 4e).

In this study, therefore, we focused on five selected GWAS analyses, including 478-DTVI-DTIA-DSIA, 4X-DSVI-MNY, 8X-DTVI-DSIA, lowland-mean TL, and upland-DTIA-MTL (Figure 4). In total, there were 27 significant peaks (one overlapped between 478 and the lowland group) across 12 chromosomes: 1K, 2K, 2N, 3K, 3N, 4N, 5K, 5N, 6N, 7K, 7N, and 9N. Accumulatively, when we considered phenotypic variance explained (PVE) by all 27 significant SNP markers for each trait, PVE of DSIA was the highest at 26.52% and PVE of DTIA was the lowest at 3.28% (Table 3).

In 478-DTVI-DTIA-DSIA, the significant markers were on chromosomes 2K, 2N, 3K, 4N, 5K, 7K, 7N, and 9N. These 14 SNP markers explained phenotypic variances for DTVI, DTIA, and DSIA of 4.25, 1.77, and 22.62%, respectively (Appendix A). In 4X-DSVI-MNY, there were three SNPs on chromosome 3K, 3N, and 7N, accumulatively explaining 9.78% of DSVI and 11.31% of MNY. In 8X-DTVI-DSIA, only one significant marker showed on chromosome 9N, explaining 0.64% of DTVI and 11.31% of MNY. In lowland-mean TL, the eight significant markers were on chromosomes 1K, 5N, 6N, 7N, and 9N, explaining 45% of mean TL. In up-DTIA-MTL, the significant markers were on chromosome 1K and 7K explaining 5.03% of DTIA and 7.77% of MTL. The PVEs of each SNP in each subgroup and each trait were reported in Appendix A.

In addition to the effect size of each SNP on each trait via PVE, the direction and inheritance of genotypic effects were shown in the format of the allele combination effect of each SNP on phenotypes (Figure 5). Of 27 SNPs, there were 11 SNPs that showed dominant effects, such as Chr05K.5450256 (Figure 5a), Chr02N.79645558, Chr07k.50656672, to name a few. Five SNPs showed overdominance, including Chr07N.61547213 (Figure 5b), Chr09N.106831822, Chr05K.90814719, Chr09N.106831822, and Chr09N17627471. Only Chr07N.48546809 showed an additive effect on DSIA as it contributes to the trait with PVE at 2.31% (Figure 5c).

When we compared these significant SNP markers from GWAS to the potential candidate genes for resistance to BLS in rice, Chr07N_48546809, on chromosome 7N from 478-DTVI-DTIA-DSIA linking with Pavir.7NG240300 overlapped with the BLAST hit from LOC_Os04g39430.1 on rice chromosome 4 from AE11005627 marker (Appendix A). Moreover, Chr07_54502620, on chromosome 7K from up-DTIA-MTL linking with Pavir.7KG238800 overlapped with the BLAST hit from LOC_Os04g43730 on rice chromosome 4 from ad04009558 marker.

## 3. Discussion

### 3.1. The Most BLS Resistant Population

Since the various phenotyping approaches yielded various resistant populations, it was challenging to determine the most resistant candidates for further breeding. Based on the field evaluation, BLUPs from two locations were zero due to zero broad-sense heritability. The different trend for BLS resistance between two locations was confirmed by the significant rank test. This re-ranking incidence suggested that the resistance was under effects between two locations [32,34]. If the mean of field scores were used, the upland ‘Shelter’ should be considered one of the most resistant populations across two locations with a mean of 0.44. The high resistance of ‘Shelter’ from field evaluation explained the reason in the prior study that it cannot be improved for resistance to BLS via recurrent phenotypic selection in seedlings. However, the one of most susceptible ‘SW787’ had a mean score of 2.33 out of 5. This suggested the low natural inoculation in 2017. When comparing each location, ‘SW803’ showed the lowest severity in NY but had the highest score in PA at 2.11. Therefore, this location variation needs to be reduced.

To lessen the GxE effect on the BLS, artificial inoculation was conducted in the laboratory. To handle the variation among sets and unbalanced design [35,36,37,38], BLUPs were used by fitting the effect from the set as a fixed effect. With the high correlation between vision and image analysis, the latter approach was recommended for the future BLS evaluation due to repeatability and reanalysis [39]. For the sake of comparison, the mean of percentage lesion from image analysis was considered. In the detached leaf approach, the lowest severity population from DTIA was ECS.6 at 8%, but the population appeared very susceptible to DSIA at 94%. These supported the low correlation among all three disease evaluation approaches. The low correlation between laboratory and field evaluations could be explained by the different disease pressure between the two conditions. The natural inoculation in the field varied by the weather condition of each year, whereas the laboratory inoculation was conducted with a high inoculum concentration to maximize the disease pressure that each genotype can respond to. Another counterintuition was the low correlation between detached leaf assay and leaf disk assay. This can be the result of the different ratios of leaf area to inoculum. The detached leaf assay used a 5-cm long whole leaf applied by spraying inoculum, whereas the leaf disk assay used only an 8-mm bored leaf section and a 2-microliter inoculum droplet on a single location. The difference between spraying and droplet was also shown in the assessment of early blight (*Alternaria solani*) in tomatoes [40].

The Qst-Fst of all phenotypes showed that none of them has gone under positive selection across five ecotypes. This neutrality was not expected as most of the resistance genes were hypothesized to be under natural selection [41,42,43,44,45]. However, neutral evolution of resistance genes like NBS-LRRs can be undergone in a relaxed selective constraint. For example, when pathogens are absent or at low frequency, the resistance genes may experience much weaker or no cost of resistance [46]. Such a low disease pressure can be noticed by means of field evaluation that the highest severity was only 2.33 from 5 (the most severe).

### 3.2. Dissecting BLS Resistance

There were only two significant SNP markers that overlap with the BLAST hits of candidate resistance genes from rice Chr07N_48546809 from the 478 GWAS and Chr07K_54502620 from upland GWAS (Table 2 and Appendix ATable S6). Chr07N_48546809 linked to Pavir.7NG240300 encoding for cytochromes P450, which involved diverse oxidation reactions and triterpene synthesis [47]. The most well-known triterpene against the fungal disease was Avenacins in oats (*Avena* spp.). Chr07K_54502620 linked to Pavir.7KG238800 encoding for EGF_CA, which has been proved to be the main structure of *ZmWAK* conferring resistance to maize head smut by controlling the galacturonan-binding for cell wall mediation [48]. In addition to genes overlapping with the BLAST hits of the resistance rice genes, Chr07N_61547213 showed a significant peak in both 478 and 4X GWAS. The SNPs linked with Pavir.7NG322400 encoding for PPR repeat, which was one of the potential resistance genes in rice. These PPR repeat families were commonly known for disease resistance genes [49]. Pavir.2NG397900 encoding GDP-fucose protein O-fucosyltransferase relating to cell wall development and disease resistance [50].

Although the rest of the significant peaks are unique within each group of GWAS, we considered the biological functions for the resistance over the fixation of the alleles in each population for a better resistance mechanism. In the full set of 478, on chromosome 2N, Pavir.2KG226200 was predicted to function as a zinc-binding RING finger. The zinc finger domains had the broad-spectrum nature of rice blast resistance gene *Pi54* classed as NBS-LRR, which played an important role in effector-triggered immunity (ETI) [51]. Moreover, this gene also encoded glycosyltransferase, which was responsible for cell wall synthesis and modification, leading to disease resistance [52]. Pavir.2NG424700 encoded for pyruvate dehydrogenase, which was a key protein in a tricarboxylic acid cycle. This suggested the high energy-intensive resistance response as happened in wheat leaf rust (*Puccinia triticina* Eriks) [53]. Pavir.2NG500000 encoded for casein kinase II. The protein itself did not confer the resistance, but it was often found as many motifs in Mlo [54], which was identified to provide broad resistance to powdery mildew in barley [55]. On chromosome 3K, Pavir.3KG329700 encoded DNA polymerase III, mainly known for DNA replications and repairing DNA damaged by hydrogen peroxide [56]. The DNA repair process was a part of the plant immune response [57], especially in the infection of *B. oryzae* that hydrogen peroxide was shown to accumulate in a leaf [23]. Pavir.3KG358400 encoded for serine aminopeptidase that conferred no resistance but played a major role in necrosis [58]. On chromosome 4N, Chr04N_47494808 was more closely linked to Pavir.4NG267200, which was predicted to have a hydrolase activity. The further gene, Pavir.4NG267500, was encoded for Villin, which was tissue-specific actin modifying protein responsible for anti-apoptotic activity suppressing necrosis [59,60]. On chromosome 5N, Pavir.5KG028600 encoded for tetratricopeptide repeat, which was one of five domains of the SGT1 resistance gene in Arabidopsis [61]. Pavir.5KG482000 encoded for ATP-binding cassette (ABC) transporter. There are many subgroups of ABC. In Arabidopsis, the mutant lacking the ABC transporter of penetration3/pleotropic drug resistance8 (PEN3/PDR8) conferred nonhost susceptibility to *Blumeria graminis*, suggesting that the ABC transporter was involved in exporting secreted toxins from the fungus and fungal suppressor from the host [62]. Moreover, in wheat, the ABC suggested the exportation of mycotoxin, conferring resistance to Fusarium head blight [63]. On chromosome 7K, Pavir.7KG196600 encoded for interleukin-1 receptor-associated kinase, which was the pathogen recognition resulting in plant response [64]. Pavir.7KG255000 encoded for oligopeptide transporter protein, which was also shown in rice QTL resistance to rice blast (*Manaporthe grisea*) [65]. Lastly, on chromosome 9N, Pavir.9NG844200 encoded for Mitochondrial Fe2+ transporter MMT1. Despite no direct report of resistance, mitochondria were proven to control redox homeostasis under stress [66].

In GWAS from 4X-DSVI-MNY, on chromosome 3K, Pavir.3KG095200 encoded for both peroxisomal membrane protein and ABC transporter. Due to the high involvement of reactive oxygen species (ROS) in the infection of *B. oryzae* [67], peroxidase played an important role in the pathogen-associated molecular pattern-triggered immunity (PTI) [68]. Additionally, Pavir.3NG101800 on chromosome 3N encoded ABC transporter.

The GWAS from 8X-DTVI-DSIA yielded only one significant SNP on chromosome 9N. Pavir.9NG061100 encoded for DNA topoisomerase and, more importantly, dirigent. Dirigent protein conferred resistance by the lignin and lignan synthesis [69]. The lignin synthesis was expected to relate to BLS resistance by strengthening cell walls [70].

In GWAS from lowland mean of two locations, on chromosome 1K, the further linked Pavir.1KG225100 encoded for MORC family-ATPases. This protein was the key component compromising the recognition of Turnip Crinkle Virus (CRT1) in Arabidopsis that was required for PTI, basal resistance, no-host resistance, and systemic acquired resistance [71]. On chromosome 5N, Pavir.5NG192000 encoded for COP9 signalosome complex. Interestingly, this signalosome complex interacts with SGT1, indicating the disease resistance [61]. Pavir.5NG248900 encoded basic leucine zipper (bZIP) transcription factor, which was a key modulator in hypersensitive response and salicylic acid [72]. Pavir.5NG476900 encoded for abscisic-acid-induced TB2/DP1, which was responsible for membrane turnover and reduced unnecessary secretion against Southern corn leaf blight by *Cochliobolus heterostrophus* (Drechs.) Drechs. in maize [73]. On chromosome 6N, Pavir.6NG083100 encoded homeobox-leucine zipper protein playing an important role in responses to abscisic acid under stress [74], transcriptional regulation for disease resistance in rice [75], and programmed cell death [76]. On chromosome 7N, Pavir.7NG091000 encoded sucrose synthase. The hexose sensing was important to activate defense-related genes as well as repression of photosynthetic genes [77]. On chromosome 9N, Pavir.9NG173600 encoded Phytochrome-interacting factor 4. The photoreceptors in plants played an essential role in hydrogen peroxide and ROS in the cell resulting in disease response [78]. Lastly, Pavir.9NG499500 encoded for alpha/beta hydrolase, which was a diversely biochemical protein family involving the activation of hydrogen peroxide [79].

In the last GWAS from upland DTIA-MTL, Chr01K_61692577 was linked to two interesting genes. First, Pavir.1KG372200 encoded Exo70 exocyst complex subunit, which was required for recognizing the avirulent effector AVR-Pii from rice blast resulting in ETI [80]. The other linked gene was Pavir.1KG372200 encoding for Nicotinamidase, which showed resistance to tomato mosaic virus [81].

In conclusion, the resistance to BLS from the different phenotyping approaches with low correlations suggested that resistance was sensitive to phenotyping methods in different conditions. Although it is difficult to standardize the phenotyping approach that can take into account all conditions, ‘SW788’, ‘SW806’, ‘SW802’, ‘SW793’, ‘SW781’, ‘SW797’, ‘SW798’, ‘SW803’, ‘SW795’, ‘SW805 should be considered as candidates for resistance. To increase the statistical power in SNP detection, the five subgroups of populations were conducted with multi-trait GWAS. The full population of 478 genotypes showed the most SNP markers that can be considered as potential candidate genes related to resistance to BLS with reasonable biological functions. Additionally, the SNP markers from the other subgroups provided a broader resistance mechanism. Nevertheless, one needs to be prudent regarding the utilization of these markers. The dependency of multi-trait GWAS to yield significant SNP peaks brought the challenge for genomic-assisted breeding for resistance to BLS.

When we considered biological functions from the GWAS regardless of subgroups and trait combinations, the defense mechanisms against BLS were complicated and multifaceted, probably controlled by multiple small-effect alleles. The defense mechanism started from basal defense via cell-wall mediated proteins and PTI via peroxidase. Then, ETI depending on NBS-LRR and PPR repeat took place. The responses to necrosis were suppressed by homeostasis and anti-apoptotic proteins. To our knowledge, this was the first research to dissect the resistance to BLS in switchgrass. With the current power of detection within the population, multi-trait GWAS can provide functional insight into the resistance within multiple subgroups of genotypes.

## 4. Materials and Methods

### 4.1. Switchgrass in the Northern Association Panel

The switchgrass used in this study has been developed for accelerating breeding progress, especially for bioenergy traits at northern latitudes [9,33]. The association panel consisted of 478 genotypes from 66 populations representing the mostly upland northern population and some southern lowland population (Appendix A). ‘Population’ in this study means a seed source with a specific origin and, in some cases, breeding history. Six to ten seeds from each population were planted into individual plants called ‘genotypes.’ The association panel was initially planted in Ithaca, NY, in 2008, and then vegetatively cloned and planted in a randomized, complete block design with three replicates in 0.9 m spaced planting in Ithaca, NY, USA and Phillipsburg, PA, USA, in 2016. No fungicide has been applied to the field.

### 4.2. Disease Evaluation and Phenotype Processing

Since the recurrent phenotypic selection for resistance to BLS cannot improve the trait from screening in the seedling stage [26], the phenotypic evaluations in this study were explored in a mature stage under different conditions. Bipolaris leaf spot has been evaluated in the fields to determine the resistance in natural incidence under different environments and in the laboratory to minimize environmental effects on the resistance. Field evaluation was conducted in the same week in both locations in August 2017. Each plant was visually evaluated by a single person to reduce the variance among persons, the severity ranging from 0 to 5 as 0 = 0% BLS, 1 = 1–10% BLS, 2 = 11–25% BLS, 3 = 26–50% 134 BLS, 4 = 51–75% BLS, and 5 = 76–100% of the leaf area of the whole plant with BLS. In addition to field evaluation, to minimize environmental effects, an artificial inoculation in the laboratory was conducted. Disease severity was evaluated by detached leaf and leaf disk assay by taking leaf samples from one replicate of genotypes in Ithaca, NY. In the detached leaf assay, three healthy leaves were randomly selected from 30 cm below the top of the shoot to control the same stage of leaf development. The leaves were cut into five centimeters, washed with deionized water, placed in a pre-wet petri dish bedded with filter paper, and kept cold in a cooler in the field, then refrigerated overnight.

The inoculum was prepared from a subculture of a single conidium isolated from a ‘Carthage’ switchgrass leaf in the warm season biofuels field experiment in Ithaca, NY (Waxman, 2011). The day after collecting the detached leaf, the three-week-old plate was flooded, filtered via gauze, and the concentration adjusted to 10^5^ conidia⋅mL^−1^ by hemocytometer with two drops of Tween-20 per 100 mL. An airbrush pressuring at ten psi was used to spray inoculum on each dry-surfaced adaxial side of the leaf on each plate. After letting the droplets of inoculum dry and firmly attached to the leaf surface, the plates were then sealed with paraffin tape to maintain high moisture conditions. Each plate contained three detached leaves from one genotype. The plates were placed on the table randomly and kept at room temperature with 12-h light for seven days showing necrotic lesions (Figure 6). In each set of the experiment, 85 to 200 genotypes were sampled with some overlapping genotypes in one to three duplicated plates and additional non-inoculated plates as controls. The leaf samplings were done weekly from May to June 2017 and controlled the similar stage of a leaf by sampling from the same height from the top. A total of seven sets of the experiment eventually covered all 478 genotypes. The disease evaluation was conducted by vision and image analysis software. Both evaluations were based on a percentage of leaves covered with lesions. The evaluation by vision was from 0 to 100% with a 10% increment. Then, each leaf was dried and scanned with a flatbed scanner (Canon CanoScan LiDE 700F manufactured by Canon Inc, Tokyo, Japan) at a resolution of 1200 dpi, and images were saved in .tiff format. The best linear unbiased predictions (BLUPs) were applied to extract random effects of the genotypes fitting the experiment factor as a fixed effect [35,36,37,38].

In the leaf disk evaluation, all 478 genotypes were collected on the same day in August 2017 by cutting the same age leaves from each genotype and keeping them refrigerated overnight. On the next day, the leaves were washed with deionized water, surface dried, bored by keeping the midrib in the middle into an 8-mm disk, and placed on a water agar plate. There were 28 leaf disks in each plate, including four check disks (control and inoculated disks of resistant ‘SW43_09’ and susceptible ‘SW122_02’) and three replicated disks of eight genotypes (Figure 6d). The inoculum was prepared the same way described above. For each leaf, 2 µL of inoculum was dropped in the middle of the right side of the leaf. After letting the droplet dry, all 64 plates were sealed with paraffin tape to keep high moisture, randomly placed on a table, and kept under 12-h light at room temperature. The disease evaluation was conducted at seven dpi both by vision and image analysis software. The percent of leaf disk covered with lesions and necrotic tissue was assessed by a vision from 0% to 100% with 10% increments. Moreover, on the same day, a photo was taken of each plate with a digital DSLR Canon Rebel T6 DSLR camera with a resolution of 2644 × 3084 in .JPG.

Images from both detached leaf and leaf disk assays were analyzed using ImageJ macro (the settings available upon request). In brief, each detached leaf or leaf disk was measured for total leaf area and total necrosis lesion area (yellow and black area). The percentage of the area covered with lesions was computed by dividing the total necrotic lesion area by the total leaf area.

Package ‘LME4’ [82] in R was used to calculate BLUPs from various disease evaluations. For the field evaluation, BLUPs were computed based on each location separately and on the two locations combined using genotypes nested in populations and replicates as random effects in each location model and populations as a fixed effect. BLUPs for the two locations combined were fitted genotype nested in population, replicates, locations, the interaction between genotypes and location, as random effects, and populations as a fixed effect. In addition to generating BLUPs from field evaluation, the highest score, which was the most severe symptoms observed from each location and two locations over replicates, were used due to successful QTLs of resistance to foliar symptoms caused by potato virus Y in autotetraploid potato (*Solanum tuberosum* L.) [83]. The highest scores were used on the assumption that in the field, there was a chance that some replicates may expose or avoid the pathogens differently. The maximum scores were considered the worst response of each genotype. For disease evaluations under laboratory conditions, since all phenotypes were expressed as percentages to leaf area, log transformation was performed before computing BLUPs. For the detached leaf assay, BLUPs were computed with experiment, genotype nested in populations, the interaction between genotypes and experiment, plates within the experiment, and leaf replicates of genotypes as random effects and populations as a fixed effect. For the leaf disk assay, BLUPs were calculated with plates as a fixed effect and genotype, replicates within plates, and genotype within plates as random effects and populations as a fixed effect. Broad-sense heritability was estimated from variances in each model, and standard error was computed by the bootstrap approach 1000 times (Appendix A). Moreover, phenotypic correlations were computed between resistance to Bipolaris leaf spot to the other 20 morphological and biomass quality traits from a previous study [33], such as plant height, anthesis date, acid detergent lignin, minerals, ethanol·g^−1^ dry weight, etc.

Therefore, the resistance to BLS was evaluated via three approaches—detached leaf, leaf disk, and field evaluation—providing phenotyped traits including BLUPs of detached leaf percent lesion via vision and image analysis (DTVI and DTIA), BLUPs of leaf disk percent lesion via vision and image analysis (DSVI and DSIA), the highest score from two locations, NY and PA (MTL, MNY, and MPA).

### 4.3. Genotyping, Linkage Disequilibrium Analysis and Population Structure

The exome-capture sequencing was conducted as previously described [84]. In short, the DNA of each genotype was processed via exome-capture using the Roche-Nimblegen exome-capture probe set [85] and DNA sequencing. The sequences were aligned to the *P. virgatum* genome assembly v.4.1 (*P. virgatum* v.4.1 [86]) for SNP discovery [87]. The reads were sorted with PicardTools version 2.1.1 [88] via functions of SortSam and MarkDuplicates. Samtools version 1.3.1 [89] was used to pile up files with base alignment quality disabled and map quality adjustment disabled. In this calling, SNPs were required to be bi-allelic, sequenced in all samples, be monomorphic in at least two samples, and filtered with a minimum read mapping quality score of 30 and more than 5X coverage in more than 95% of the samples. At each SNP, the genotype dosages range from zero to two copies of minor alleles and can be nonintegers by using the EM algorithm [90]. Naturally, the switchgrass association panel includes allopolyploid switchgrass: tetraploids (4×), octoploids (8x), and hexaploids (6x). It was challenging to perform GWAS with polyploid models. In this study, we modeled them under the assumption of disomic inheritance, similar to the study of GWAS of flowering time in the previous study [29]. This was because tetraploid switchgrass was confirmed to have disomic inheritance [91]. Although octaploid switchgrass had four copies of each homologous chromosome, which made it difficult to precisely determine heterozygous genotypes, the disomic segregation was used for the model with the caution of the increasing standard error of estimates in GWAS. Moreover, linkage disequilibrium (LD) was computed via the ‘-r2’ tag in PLINK as r^2^ between all SNPs within 1 MB and recorded all values [92]. Principle component analysis was used to evaluate the population structure via PCA in the FactoMineR package [93]. Additionally, ADMIXTURE was used to evaluate population structure with five-fold cross-validation [94].

We were also interested in testing whether the variations in phenotypes were from the high differentiation across five ecotypes (lowland north, lowland south, upland east, upland north, upland west) based on Qst-Fst analysis. Estimated Qst-Fsts were computed by R package “QstFstComp” by bootstrapping 1000 times by sampling random genotypes [95].

### 4.4. Candidate Genes Based on Resistance in Rice

Since *B. oryzae* is one of the major pathogens in rice (*Oryza sativa*), major resistant QTLs have been identified in recombinant inbred lines (RILs) and doubled haploids (DH) using restriction fragment length polymorphism (RFLP), simple sequence repeat (SSR) markers, and sequence-tagged site (STS) [30,31,94]. Three major QTLs in Chromosomes 1, 4, and 11 were then studied further in near-isogenic lines (NILs) and 13 SNP markers linked with the resistance [30]. The NILs are BC3F5 between indica ‘Tadukan’ (resistance) and temperate japonica ‘Koshihikari’ (susceptible). According to the previous GWAS in rice, linkage disequilibrium (LD) was ~100 kb in indica and ~200 kb temperate in japonica [96]. Thus, in this study, the candidate resistance genes were screened from 200 kb around the 13 SNPs from *Oryza sativa* genome assembly v.7.0 (*Oryza sativa* v.7.0 [86]). The candidate genes from rice are listed in Appendix A. The sequences of these candidate genes were used to identify potential homologs in switchgrass using the BLASTP tool on Phytozome and kept the hits with E-value < 10^−10^ [86].

### 4.5. Genome-Wide Association Studies

Genome-wide Efficient Mixed Model Association (GEMMA) [97] was used to implement a multivariate linear mixed model for GWAS of single- and multiple-phenotyping approaches for resistance to BLS to analyze each phenotype and combined phenotypes for improving statistical power [98,99]. Kinship was included as a random effect. Moreover, the first three principal components (PCs) were included as fixed covariates. The GWAS was conducted in five groups, including all 478 genotypes, tetraploid (4×), octaploid (8X), lowland, and upland genotypes, to determine if there were any SNPs linked to specific switchgrass population. Minor allele frequency (MAF) at 0.05 was used to filter the minor alleles. In total, there are 135,791 SNP markers used in association analysis. To correct multiple testing, the false discovery rate (FDR) was calculated as
(1)$$\mathrm{FDR}=\frac{\frac{P}{k}\left(1-\frac{A}{T}\right)}{\frac{A}{T}\left(1-\frac{P}{k}\right)}$$
where P is the *p*-value tested at 0.1; k is the number of traits in the combination; A is the number of SNP that were significant at the *p*-value tested; T is the total number of SNP tested. The significant markers then were used to determine candidate genes linked to them within the range of LD and search in JBrowse in *P. virgatum* v.4.1 in Phytozome (e.g., if the position of the marker was 4019500 on chromosome 9K and LD was 20 kb, “Chr09K:4019500..4039500” was used). In case the candidate gene was not fully aligned within that LD range, we considered the gene as a candidate if 80% of its length was within the LD range. Since some of the SNP positions could not be aligned to the *P. virgatum* genome assembly v.4.1, they were excluded from determining the candidate genes.

## Figures and Tables

**Figure 1 plants-11-01362-f001:**
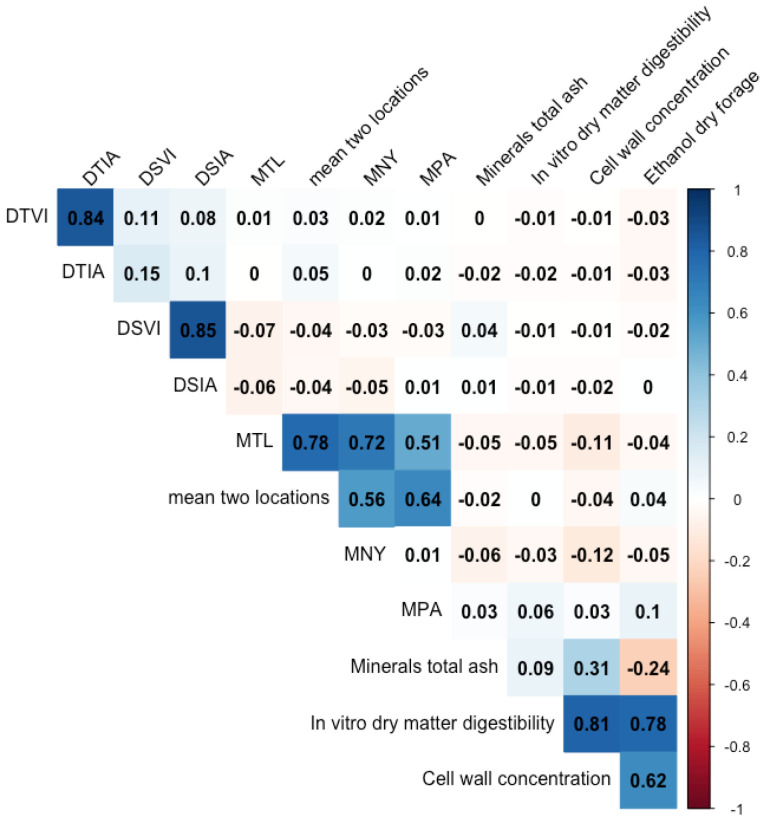
Correlation plot among BLUPs of severity from the detached leaf via vision (DTVI), via image analysis (DTIA), from leaf disk assay via vision (DSVI), via image analysis (DSIA), the highest score between two locations (MTL), mean from the two locations (TL), minerals total ash, in vitro dry matter digestibility, cell wall concentration, and ethanol conversion per day from a previous study [33].

**Figure 2 plants-11-01362-f002:**
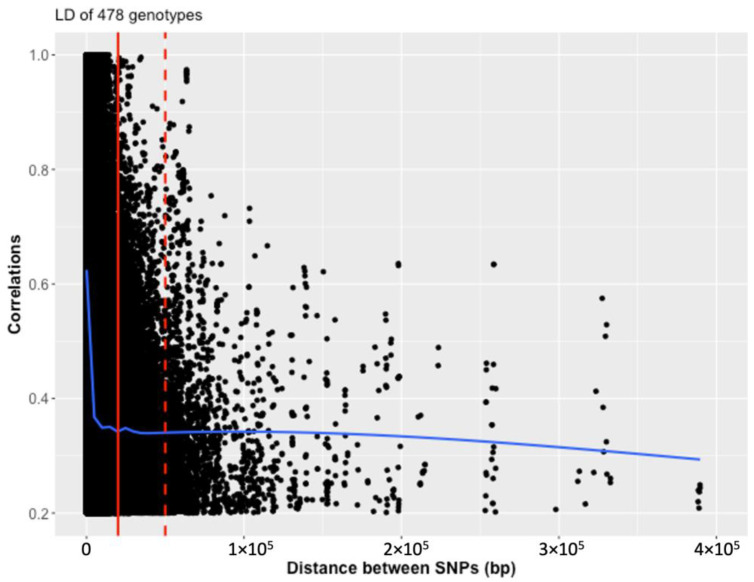
Scatter plot showing the linkage disequilibrium (LD) decay by plotting physical distance in base pairs against the LD estimate as correlations (R^2^) in 478 genotypes. The LD (blue curve) decays rapidly within 20 kb (solid red line) and reaches background levels around 50 kb (dashed red line).

**Figure 3 plants-11-01362-f003:**
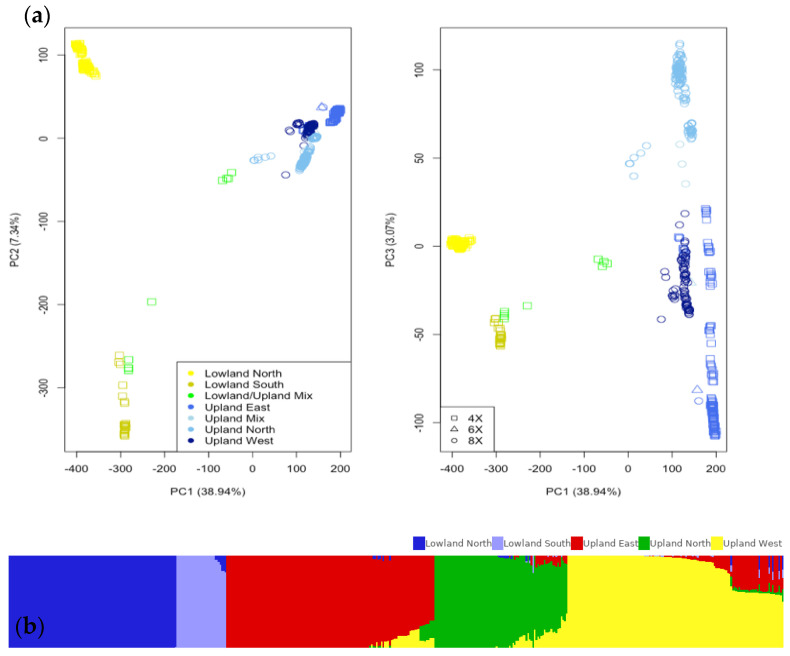
(**a**) The 478 switchgrass genotypes from the Northern Association Panel showed the different distributions based on ecotype and region from principal component analysis (PC). Colors represented different ecotypes, and shapes represented different ploidy levels. Percentage variances explained by PCs were in parentheses. The lowland ecotype can be differentiated into two regions from PC1 and PC2, and the upland ecotype can be grouped into three regions from PC1 and PC3. (**b**) Admixture also showed the separations of the genotypes into five gene pools.

**Figure 4 plants-11-01362-f004:**
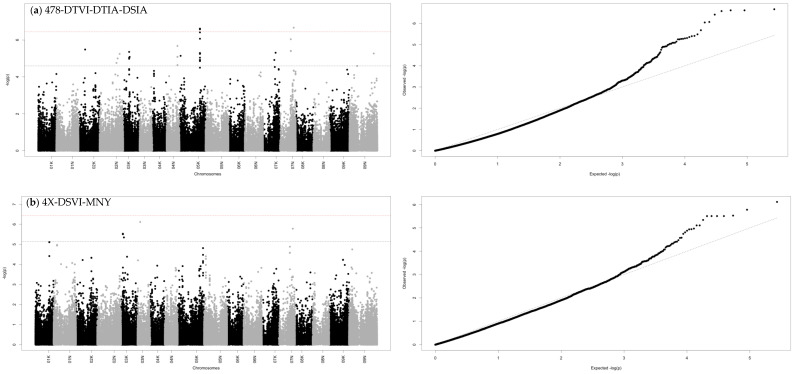
(**Left**) Manhattan plot showed genetic association of (**a**) DTVI-DTIA-DSIA in 478 genotypes, (**b**) DSVI-MNY in 4X genotypes, (**c**) DTVI-DSIA in 8X genotypes, (**d**) mean TL in lowland genotypes, (**e**) DTIA-MTL in upland genotypes. The black dashed line represented the FDR threshold (0.1), and the red dashed line represented the Bonferroni correction threshold. On the x-axis, the physical positions of the SNPs were aligned in 18 chromosomes of *P. virgatum.* (Right) Quantile-quantile (QQ) plots between the distributions of observed to expected *p*-values for GWAS of each trait combination in each genotype group. Abbreviations: BLUPs of severity from the detached leaf via vision (DTVI), BLUPs of severity from the detached leaf via image analysis (DTIA), BLUPs of severity from the leaf disk assay via vision (DSVI), BLUPs of severity from the leaf disk assay via image analysis (DSIA), mean of field evaluation of BLS in two locations (mean TL), the highest score of BLS in NY (MNY), and the highest score of BLS in the two locations (MTL).

**Figure 5 plants-11-01362-f005:**
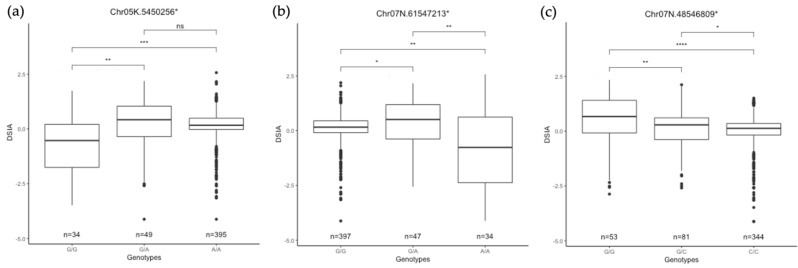
Examples of changes in DSIA associated with alleles at selected candidate SNP among 478 genotypes. The “*” behind the SNP names mean *P. virgatum* genes that overlap with the BLAST hits from BLS resistance genes from *Oryza sativa*. (**a**) Chr05K.5450256 showed a dominant effect. (**b**) Chr07N.61547213 showed an overdominant effect. (**c**) Chr07N.48546809 showed an additive effect. The effects of three different allele combinations were tested at alpha = 0.05. “ns” means non-significant, “*” means significant at alpha 0.05, “**” means significant at alpha 0.01, “***” means significant at alpha 0.001, and “****” means significant at alpha 0.0001.

**Figure 6 plants-11-01362-f006:**
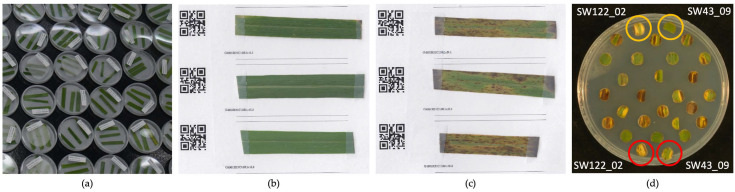
Detached leaf assay and leaf disk assay; (**a**) sealed plates of detached leaves under 12-h light 25 °C; (**b**) control inoculated and (**c**) inoculated leaves at seven dpi were taped on white paper with QR code label for scanning for image analysis. (**d**) A plate example of leaf disk assay at seven dpi. Each plate consisted of 28 leaf disks. SW43_09 was used as a resistant check and SW122_02 as a susceptible check. Two control leaf disks (yellow circles) and two inoculated leaf disks (red circles) were included in all plates. The rest of the leaf disks (24) consisted of eight genotypes with three replicates placed in a consecutive row from left to right.

**Table 1 plants-11-01362-t001:** Broad-sense heritability of each trait in each group.

Traits	H^2^ ± s.e.				
	478 Genotypes	4X	8X	Lowland	Upland
Severity from the detached leaf via vision (DTVI)	0.76 ± 0.02	0.84 ± 0.04	0.66 ± 0.06	0.87 ± 0.03	0.71 ± 0.05
Severity from the detached leaf via image analysis (DTIA)	0.74 ± 0.06	0.78 ± 0.05	0.72 ± 0.03	0.81 ± 0.06	0.68 ± 0.02
Severity from the leaf disk assay via vision (DSVI)	0.76 ± 0.05	0.70 ± 0.02	0.74 ± 0.05	0.35 ± 0.04	0.63 ± 0.04
Severity from the leaf disk assay via image analysis (DSIA)	0.5 ± 0.04	0.33 ± 0.06	0.52 ± 0.04	0.45 ± 0.05	0.14 ± 0.03
Severity in the field of the two locations	0.003 ± 0.06	0.003 ± 0.04	0.1 ± 0.05	0.001 ± 0.08	0.08 ± 0.06
Severity in field from NY	0.32 ± 0.05	0.31 ± 0.05	0.29 ± 0.04	0.38 ± 0.03	0.29 ± 0.04
Severity in field from PA	0.61 ± 0.04	0.68 ± 0.03	0.52 ± 0.02	0.78 ± 0.05	0.51 ± 0.03

**Table 2 plants-11-01362-t002:** Markers significantly associated with resistance to Bipolaris leaf spot in five GWAS analyses.

Cases	SNP	Chromosomes	*p*-Value	Distance from Genes (bp)	*P. Virgatum* Gene	Orthologous Genes	Identity	Predicted Functions
478-DTVI-DTIA-DSIA	Chr02K_29538993	Chr02K	3.30 × 10^−6^	1083	Pavir.2KG226200	Pahal.B01583.1 ^Pha^	99.6	Zinc-binding finger, Glycosyl transferase, cellulose biosynthesis
	Chr02N_74508772	Chr02N	1.76 × 10^−5^	899	Pavir.2NG397900	Pahal.B03217.1 ^Pha^	97.2	GDP-fucose protein O-fucosyltransferase
	Chr02N_79645558	Chr02N	1.00 × 10^−5^	1496	Pavir.2NG424700	Sobic.002G259700.1 ^Sbi^	98.4	Pyruvate dehydrogenase (acetyl-transferring)
	Chr02N_88427045	Chr02N	5.70 × 10^−6^	1046	Pavir.2NG500000	Pahal.B04009.1 ^Pha^	100	Casein kinase II, beta subunit
	Chr03K_26598614	Chr03K	9.25 × 10^−6^	1239	Pavir.3KG329700	Sevir.3G174000.1 ^Svi^	90.6	DNA polymerase III
	Chr03K_28867984	Chr03K	8.31 × 10^−6^	788	Pavir.3KG358400	Sevir.3G185400.1 ^Svi^	93.3	Serine aminopeptidase
	Chr04N_47494808	Chr04N	8.29 × 10^−6^	235	Pavir.4NG267200	Sobic.010G215800.1 ^Sbi^	78.2	OSIGBa0118P15.3 relating to protein hydrolase activity
				8698	Pavir.4NG267500	LOC_Os06g44890.1 ^Osa^	93.3	Villin as an anti-apoptotic protein
	Chr05K_5450256	Chr05K	7.26 × 10^−6^	47	Pavir.5KG028600	Pahal.E04447.1 ^Pha^	97.5	Tetratricopeptide repeat
	Chr05K_90814719	Chr05K	8.62 × 10^−7^	321	Pavir.5KG482000	Zm00008a014067_T0 ^Zma^	94.9	ABC transporter
	Chr07K_50656672	Chr07K	1.22 × 10^−5^	207	Pavir.7KG196600	Zm00008a007144_T01 ^Zma^	95.2	Interleukin-1 receptor-associated kinase
	Chr07K_56339702	Chr07K	4.94 × 10^−6^	756	Pavir.7KG255200	Pahal.G01334.1 ^Pha^	92.1	Uncharacterized ACR, COG1399
				930	Pavir.7KG255000	LOC_Os04g44320.1 ^Osa^	84.2	Oligopeptide transporter protein
	Chr07N_48546809	Chr07N	9.13 × 10^−7^	524	Pavir.7NG240300 *	LOC_Os04g39430.1 ^Osa^	88.1	Cytochrome P450 CYP4/CYP19/CYP26 subfamilies
	Chr07N_61547213	Chr07N	2.18 × 10^−7^	571	Pavir.7NG322400	LOC_Os04g51350.1 ^Osa^	91.1	PPR repeat family
	Chr09N_106831822	Chr09N	5.42 × 10^−6^	20	Pavir.9NG844100	Sobic.001G445100.1 ^Sbi^	73.5	Putative uncharacterized protein
				9902	Pavir.9NG844200	GRMZM2G118497_T0 ^Zma^	98.5	Mitochondrial Fe2+ transporter MMT1 and metal tolerate
4X-DSVI-MNY	Chr03K_8647071	Chr03K	3.12 × 10^−6^	733	Pavir.3KG095300	Sobic.009G005500.1 ^Sbi^	93.6	Histone acetyltransferases subunit 3
				6957	Pavir.3KG095200	Sobic.009G005400.1 ^Sbi^	96.8	Peroxisomal membrane protein, ABC transporter
	Chr03N_12119852	Chr03N	7.69 × 10^−7^	2038	Pavir.3NG101800	LOC_Os12g22284.1 ^Osa^	89.3	ABC-2 type transporter
	Chr07N_61547213	Chr07N	1.66 × 10^−6^	571	Pavir.7NG322400	LOC_Os04g51350.1 ^Osa^	91.1	PPR repeat family
8X-DTVI-DSIA	Chr09N_3891068	Chr09N	2.66 × 10^−7^	535	Pavir.9NG061100	Sobic.001G040600.1 ^Sbi^	94.3	DNA topoisomerase (ATP-hydrolyzing), dirigent
lowland-mean TL	Chr01K_36400224	Chr01K	3.08 × 10^−6^	1354	Pavir.1KG225400	Sobic.004G141800.2 ^Sbi^	96	Similar to Os02g0469200 protein
				10,167	Pavir.1KG225100	Sevir.1G159400.1 ^Svi^	88	MORC family ATPases
	Chr05N_24540124	Chr05N	5.43 × 10^−6^	1020	Pavir.5NG192000	Sevir.5G038000.1 ^Svi^	100	COP9 signalosome complex
	Chr05N_35504012	Chr05N	2.47 × 10^−6^	360	Pavir.5NG248900	Sevir.5G193800.1 ^Svi^	88.5	Basic leucine zipper (bZIP) transcription factor
	Chr05N_84742282	Chr05N	5.20 × 10^−6^	1309	Pavir.5NG476900	Zm00008a032406_T01 ^Zma^	71.5	TB2/DP1, HVA22 family
	Chr06N_14870066	Chr06N	2.40 × 10^−6^	1766	Pavir.6NG083100	Sevir.6G037200.1 ^Svi^	97.6	Homeobox-leucine zipper protein
	Chr07N_22055455	Chr07N	1.89 × 10^−5^	1088	Pavir.7NG091000	Sobic.010G276700.1 ^Sbi^	98.2	Sucrose synthase
	Chr09N_17627471	Chr09N	1.47 × 10^−5^	2258	Pavir.9NG173600	Pahal.I02469.1 ^Pha^	92.7	Phytochrome-interacting factor 4
	Chr09N_82637734	Chr09N	1.16 × 10^−6^	1030	Pavir.9NG499500	Zm00008a001876_T01 ^Zma^	95.3	Alpha/beta hydrolase family
upland-DTIA-MTL	Chr01K_61692577	Chr01K	1.52 × 10^−6^	6831	Pavir.1KG372200	Sobic.010G272700.1 ^Sbi^	53.6	Exo70 exocyst complex subunit
				2646	Pavir.1KG372200	Zm00008a022268_T01 ^Zma^	95.9	Nicotinamidase
	Chr07K_54502620	Chr07K	4.48 × 10^−6^	3810	Pavir.7KG238800 *	LOC_Os04g43730.1 ^Osa^	73.6	Calcium-binding EGF domain (EGF_CA), Wall-associated receptor kinase galacturonan-binding

* *P. virgatum* genes that overlap with the BLAST hits from BLS resistance genes from *Oryza sativa*, ^Osa^ gene from *Oryza sativa*, ^Pha^ gene from *Panicum hallii*, ^Sbi^ gene from *Sorghum bicolor*, ^Svi^ gene from *Setaria viridis*, ^Zma^ gene from *Zea may.*

**Table 3 plants-11-01362-t003:** The phenotypic variance explained (PVE) (adjusted PVE) by accumulative 27 significant SNP markers across all analyses for each approach.

Traits	PVE (Adjusted PVE)
DTVI	11.68 (6.38)
DTIA	8.76 (3.28)
DSVI	25.03 (20.53)
DSIA	30.68 (26.52)
mean TL	15.73 (10.67)
MNY	15.74 (10.68)
MTL	13.7 (8.52)

## Data Availability

Not applicable.

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
