# Peer review of "Genome-Wide Associations with Resistance to Bipolaris Leaf Spot (Bipolaris oryzae (Breda de Haan) Shoemaker) in a Northern Switchgrass Population (Panicum virgatum L.)"

_plants, 2022, doi:10.3390/plants11101362_

Round 1
Reviewer 1 Report
In this article, the authors did Genome-Wide Associations with Resistance to Bipolaris Leaf Spot (Bipolaris oryzae (Breda de Haan) Shoemaker) in a Northern Switchgrass Population (Panicum virgatum L.). In this study, authors determine the resistant populations via multiple phenotyping approaches and identify potential resistance genes from genome-wide association studies (GWAS) in the switchgrass northern association panel. The disease resistance was evaluated from both natural (field evaluations in New York and Philadelphia) and artificial inoculations (detached leaf and leaf disk assays). The most resistant populations based on a combination of three phenotyping approaches – detached leaf, leaf disk, and mean from two locations – were ‘SW788’, ‘SW806’, ‘SW802’, ‘SW793’, ‘SW781’, ‘SW797’, ‘SW798’, ‘SW803’, ‘SW795’, ‘SW805’. The GWAS from the association panel showed 27 significant SNPs on 12 chromosomes: 1K, 2K, 2N, 3K, 3N, 4N, 5K, 5N, 6N, 7K, 7N, and 9N. These markers accumulatively explained the phenotypic variance of the resistance ranging from 3.28-26.52%. Within linkage disequilibrium of 20 kb, these SNP markers linked with the potential resistance genes included the genes encoding for NBS-LRR, PPR, cell-wall related proteins, homeostatic proteins, and anti-apoptotic proteins, and ABC transporter.
The manuscript is very well written, and I found no major flow in this study and hence recommend it for publication in its current format.
Author Response
I deeply appreciate your contribution to reviewing the manuscript. I hope this research can provide important information for the plant breeding community.
Best regards,
Kittikun Songsomboon
Reviewer 2 Report
The manuscript by Songsomboon et al. is a great contribution to understanding the genetic basis of resistance to bipolaris leaf spot in switchgrass. I found the manuscript well written and the results presented deserve publication in Plants. However, I would like to strongly suggest that the manuscript be shortened appropriately by presenting only the major and significant results of this study and removing some additional figures and tables that do not need to be included even as supplementary materials. In addition, some corrections are suggested as follows:
1- Please remove "Histogram" from the legend of Figures S1-S2.
2- Line 213: I think table S4 is the correct table here.
3- Line 221: It is better to put this paragraph before paragraph 2.3.
4- Line 511: "To our knowledge" is correct.
The figures which the authors may consider to be not completely necessary are the following:
1- Figures S5-S19 2- Tables stat1-stat17 3- page 1-11 from the file: Songsomboon_supplement_file_plant. pdfAuthor Response
Please see the attachment.
